# Diagnosis of Chronic Musculoskeletal Pain by Using Functional Near-Infrared Spectroscopy and Machine Learning

**DOI:** 10.3390/bioengineering10060669

**Published:** 2023-06-01

**Authors:** Xinglin Zeng, Wen Tang, Jiajia Yang, Xiange Lin, Meng Du, Xueli Chen, Zhen Yuan, Zhou Zhang, Zhiyi Chen

**Affiliations:** 1Institute of Medical Imaging, Hengyang Medical School, University of South China, Hengyang 421000, China; 2Faculty of Health Sciences, University of Macau, Macau SAR, China; 3Centre for Cognitive and Brain Sciences, University of Macau, Macau SAR, China; 4Department of Rehabilitation Medicine, The First Affiliated Hospital, Sun Yat-sen University, Guangzhou 510000, China; 5School of Life Science and Technology, Xidian University, 266 Xinglong Section of Xifeng Road, Xi’an 710126, China

**Keywords:** chronic pain, fNIRS, machine learning, graph theory

## Abstract

Chronic pain (CP) has been found to cause significant alternations of the brain’s structure and function due to changes in pain processing and disrupted cognitive functions, including with respect to the prefrontal cortex (PFC). However, until now, no studies have used a wearable, low-cost neuroimaging tool capable of performing functional near-infrared spectroscopy (fNIRS) to explore the functional alternations of the PFC and thus automatically achieve a clinical diagnosis of CP. In this case-control study, the pain characteristics of 19 chronic pain patients and 32 healthy controls were measured using fNIRS. Functional connectivity (FC), FC in the PFC, and spontaneous brain activity of the PFC were examined in the CP patients and compared to those of healthy controls (HCs). Then, leave-one-out cross-validation and machine learning algorithms were used to automatically achieve a diagnosis corresponding to a CP patient or an HC. The current study found significantly weaker FC, notably higher small-worldness properties of FC, and increased spontaneous brain activity during resting state within the PFC. Additionally, the resting-state fNIRS measurements exhibited excellent performance in identifying the chronic pain patients via supervised machine learning, achieving F1 score of 0.8229 using only seven features. It is expected that potential FC features can be identified, which can thus serve as a neural marker for the detection of CP using machine learning algorithms. Therefore, the present study will open a new avenue for the diagnosis of chronic musculoskeletal pain by using fNIRS and machine learning techniques.

## 1. Introduction

Chronic pain (CP) refers to pain that persists after a normal healing period, lasting or recurring for over 3 months [1,2]. CP affects approximately 20% of the population worldwide, demonstrating a myriad of biomedical, psychosocial, and behavioral disturbances [3]. In particular, CP patients are more vulnerable to developing emotional and cognitive disorders. According to previous studies (Barnett, Mercer [4]), 20–50% of CP patients might develop co-morbid depression and cognitive disorders such as dysfunctions in executive function, decision making, and social cognition [5]. Meanwhile, CP patients might also exhibit significant brain structural and functional alterations due to changes in pain processing and disrupted cognitive functions [6,7]. Presently, a reliable neurophysiological measure for objectifying pain is still lacking [8]. Interestingly, due to the role of the prefrontal cortex (PFC) in pain processing and regulating high functions, the functional connectivity (FC) of the PFC might be a promising measure for the diagnosis of CP [9]. For example, Ihara, Wakaizumi [10] discovered that CP patients exhibited significantly different functional brain networks in the PFC compared to pain-free controls. Moreover, the degree of altered FC between the nuclear accumbens and the medial PFC was significantly correlated with pain chronicity [11]. In addition to the FC changes of the PFC, studies have revealed impaired topographical properties of FC associated with CP via graph theory analysis [12,13].

More importantly, recent advances in neuroimaging techniques and methods such as electroencephalography (EEG), functional magnetic resonance (fMRI), and functional near-infrared spectroscopy (fNIRS) have offered better opportunities to fully understand the cognitive neural mechanisms underlying CP during a task [14] or at rest [15]. In addition, fNIRS is a wearable-conducive, low-cost, and noninvasive neuroimaging technique that measures the concentration changes of oxygenated (HbO) and deoxygenated (HbR) hemoglobin in brain tissue following neuronal activity [16,17]. Unlike EEG and fMRI, fNIRS can be carried out in a natural environment, which is not very sensitive to motion artifacts [18]. Further, fNIRS data combined with machine learning can assist the medical and clinical diagnosis of various psychiatric and neurological disorders, such as Alzheimer’s disease, Parkinson’s disease, post-neurosurgery dysfunction, anxiety disorders, and childhood disorders [19]. For example, Yang and Hong [20] utilized fNIRS and a pre-trained convolutional neural network model to analyze the difference in FC between mild cognitive impairment patients and healthy controls (HCs). Xu, Liu [21] used fNIRS and a deep learning model to examine the potential patterns of temporal variation in the resting state in patients with autism spectrum disorder, achieving a high classification accuracy of 95.7%. These studies demonstrated the promising potential of fNIRS-based machine learning in detecting and classifying disorders and predicting their severity. To the best of our knowledge, no work has been conducted using fNIRS and machine learning methods to reveal the unique FC patterns for the detection of CP.

In this study, we will analyze whether FC and spontaneous brain activity of the PFC are changed in CP patients compared to those of HCs. We will also examine the topological properties of the PFC, such as the clustering coefficient and path length, by using graph theoretical network analysis [22]. It is expected that potential FC features can be identified, which can serve as a neural marker for the detection of CP using machine learning algorithms. Therefore, the present study will open a new avenue for the diagnosis of chronic musculoskeletal pain through fNIRS and machine learning techniques.

## 2. Methods

### 2.1. Participants

In this study, 52 right-handed participants, including 19 chronic musculoskeletal pain patients (25.3 ± 4.6 years, 12 females) and 32 age- and gender-matched HCs (24.7 ± 4.2 years, 18 females), were recruited from the First Affiliated Hospital of Sun Yat-sen University. All patients, who were afflicted with pain that had lasted over 6 months and free of any neurologic or metabolism diseases, were diagnosed with chronic musculoskeletal pain by three physicians. All participants provided informed consent prior to the experiment. The present study protocol was approved by the Institutional Review Board at both the University of Macau and Sun Yat-sen University.

### 2.2. fNIRS Data Acquisition and Preprocessing

All participants underwent a 5 min session of resting-state fNIRS recordings. They were seated in a comfortable chair and instructed to keep still and not to think about anything purposely. It was observed that none of the participants slept during the fNIRS data recording. Our experiment was conducted by utilizing a fNIRS system (Oxymon Mk III, Artinis, The Netherlands) transmitting at two wavelengths, namely, 760 and 850 nm, to measure HbO and HbR concentration changes with 50 HZ sampling rate (Figure 1). The fNIRS system consisted of 2 near-infrared light source emitters and 8 detectors, yielding 8 channels in total. The distance between each source and detector pair was set to 3 cm. In addition, the Montreal Neurological Institute (MNI) coordinates of each fNIRS channel were measured using an ICBM-152 head model, which was based on the international 10–20 system for EEG recording. Then, the coordinates were processed using NIRS-SPM to estimate the MNI coordinates and associated brain regions of the optodes and channels together with the probability of the channels (Table 1). This probability measure describes how well the estimated MNI coordinates accurately correspond to specific brain regions.

The fNIRS data were preprocessed using NIRS-KIT (http://www.nitrc.org/projects/nirskit/ accessed on 9 December 2022). The recordings from the first and last 15 s were excluded due to potential body movements. We then used detrending and the temporal derivative distribution repair method to reduce data drift and correct artificial motions, respectively [23]. To minimize physiological noise due to heart pulsation (1~1.5 Hz), respiration (0.2~0.5 Hz), and blood pressure (Mayer) waves (~0.1 Hz), the data were further filtered with a bandpass of 0.01–0.1 Hz [24]. In this study, only HbO signals were analyzed since they exhibited more sensitive changes to regional cerebral blood flow [25].

### 2.3. Brain Network Analysis

fNIRS channels and connections between them were defined as nodes and edges, respectively. Pearson correlation coefficient between the time courses of each pair of channels was calculated to construct individual-level brain networks, generating an 8 by 8 connection matrix for each participant. Fisher *r*-to-t method was used to convert the correlation coefficients to *t* values [26]. Two sample tests were applied to determine the altered connectivity networks in CP patients compared to those of HCs while correcting for the false discovery rate (FDR) [27].

### 2.4. Analysis of the Amplitude of Low-Frequency Fluctuations

Previous studies demonstrated that spontaneous neural activity during rest were particularly correlated with low-frequency blood-oxygen-level-dependent (BOLD) signals, which can be represented by the amplitude of low-frequency fluctuations (ALFF) [28]. In particular, significantly altered ALFF values were identified in the PFC at rest or while performing a task among CP patients compared to HCs [28,29,30]. These functional abnormalities in the PFC have been revealed in resting state fMRI, which is now extended to the fNIRS field. Therefore, to characterize spontaneous brain activity, ALFF was calculated as the averaged amplitude within 0.01–0.1 Hz for fNIRS data [31]. According to Fourier transform, the time series of each channel was converted into the amplitude spectrum in the frequency domain. Then, to increase the normalization of fNIRS data distribution, the standard ALFF (zALFF) was obtained by subtracting the global mean value across all channels, which was then divided by the standard deviation across all channels [26].

### 2.5. Graph Theory Analysis

Graph theory measurements were generated using the GRETNA toolbox (https://www.nitrc.org/projects/gretna/ accessed on 12 December 2022). The network organizations were assessed through defined sparsity, which is the number of existing edges divided by the maximum possible number of edges within a network [32]. The selected range of the sparsity threshold was from 0.2 to 1 (interval = 0.01) due to the small-worldness of human brain networks [33]. For each subject at each time-scanning duration, binarized adjacency networks were generated by using these chosen thresholds. Once the graphs were constructed, the properties of the graphs were inspected. Compared with random networks, small-world networks have similar characteristic path lengths but higher characteristic clustering. The small-world index was calculated as follows:(1)$$\sigma =\frac{C/Crand}{L/Lrand}$$

When the value of σ was larger than 1, the network was defined as possessing small-world characteristics [33]. The graph measurements can be categorized as global measurements and local node measurements, where each graph has only one value as a global measure and eight node values for local measures. The global measurements, including global and local efficiency, assortativity, synchronization, and hierarchy, were examined to reveal the properties of functional segregation, functional integration of information flows within the brain network, and network resilience against failure. By contrast, the local nodal measures (i.e., clustering coefficient, shortest path length, local efficiency, degree centrality, betweenness centrality, community index, and participant coefficient) were computed to investigate the properties of 8 putative functional areas of brain [33]. These measures then generated the final feature vector for each chronic pain patient and healthy control. Local metrics were calculated for each regional node to identify the most important nodes during graph analysis at a connection density of 43%, which has shown the best ability to differentiate CP patients from HCs with small world properties [34].

### 2.6. Feature Selection and Classification

A large number of features might cause overfitting, so the number of √n values with highly correlated features is generally used, with n referring to the sample size [35]. Feature selection module can select an optimal subset of features from the original feature set, which is a required step for high-dimensional data (such as fMRI). Here, we adopted the Fisher score feature selection algorithm, which is a univariate feature selection algorithm. It is independent of the class distribution when applied to determine the discriminatory power of individual features between two classes of equal probability [36]. Fisher score for each feature in a two-class problem is defined as:(2)$$FS=\frac{n_{1}{\left(m_{1}-m\right)}^{2}+n_{2}{\left(m_{2}-m\right)}^{2}}{n_{1}\sigma_{1}+n_{2}\sigma_{2}}$$
in which *m* is the mean of the feature, *m*_1_ and *m*_2_ are feature mean values of each class, *σ*_1_ and *σ*_2_ are respective variances, and *n*_1_ and *n*_2_ are the numbers of samples in the classes.

Thus, according to the sample size, seven features with the highest Fisher scores from the graph theory measurements and ALFF results were selected to classify the samples, in which two were global measurements (network efficiency and clustering coefficient) and five were node measurements (local efficient of channel 5, cluster efficient of channel 5, local efficient of channel 8, community index of channel 7, and efficient of channel 7) (Table 2).

After the feature selection stage, three well-established supervised machine learning methods were used to construct the classifier. The supervised machine learning algorithms were trained by using a set of input data to produce the desired output. The supervised machine learning algorithms used in this study were linear support vector machines (SVMs), which determined a linear maximum-margin hyperplane to maximize separation between groups. All the machine learning algorithms were implemented in MATLAB (The Math Works, Natwick, MA, USA), and SVM was conducted using LIBSVM (http://www.csie.ntu.edu.tw/~cjlin/libsvm/, accessed on 20 December 2022). Linear regression was used to model the relationship between a dependent variable and one or more independent variables, whose goal is to find the best-fitting line that minimizes the distance between the predicted values and the actual values of the dependent variable. Naive Bayes works by using Bayes’ theorem to calculate the probability of each class label given the values of the input features. Naive Bayes states that the probability of a label given some evidence (the input features) is proportional to the probability of the evidence given the hypothesis multiplied by the prior probability of the hypothesis. Since the small number of participants in this study might undermine the generalizability of the classifier, we employed leave-one-out cross-validation (LOOCV). LOOCV was repeated until data from all participants had been used as the test sample once. The classification performance values from all the repetitions were averaged to obtain the final result. Accuracy was defined as measuring the proportion of correctly classified instances over the total number of instances, and another four metrics, namely, sensitivity, specificity, F1 score, and area-under-the-curve (AUC) value, were used to comprehensively evaluate the performance of the classifier in discriminating HC and CP groups.

## 3. Results

### 3.1. Correlation Matrix

By using resting-state fNIRS data, a group-level correlation matrix was generated for both the CP group and HC group, which was calculated using the mean value of the individual-level correlation matrix. Then, the network strengths between two groups were compared using an independent t-test and corrected. Group-level connectivity differences between the HC and CP groups are shown in Figure 2. Group-level analysis demonstrated that the resting-state FC strength was significantly higher (p_FDR_  <  0.05) between channels 5 and 4 for the HCs.

### 3.2. ALFF Results

As demonstrated in Figure 3, compared to the HCs, the CP patients exhibited significantly increased zALFF values in channel 8 after FDR correction (HC = −0.2495 ± 0.8044, chronic pain = 0.5176 ± 1.0098, *t* = −2.9683, *p* = 0.0047, Cohen’s d = 0.8403, power = 0.806, and 93.962% Dorsolateral prefrontal cortex (DLPFC)). However, no significantly decreased zALFF values were detected for the patient group. The findings demonstrate that the brain activation in the PFC of the CP patients was significantly higher than that in the HCs.

### 3.3. Comparison of Functional Network Characteristics

The measures of small-worldness (σ) and other functional network characteristics were calculated for both the CP and HC groups, respectively, and are provided in Figure 4 as a function of sparsity (i.e., threshold *T*). It was discovered that the σ values of the constructed brain functional networks for both groups were above 1 (Figure 4), but the CP patients showed stronger small-worldness. More importantly, among the five global measurements, we discovered that the CP group presented significantly lower global efficiency and synchronizations than the HC group.

### 3.4. Classification Results

Three machine learning algorithms were tested with respect to their ability to classify CP and HC groups. As shown in Table 3, the SVM achieved an accuracy of 75.59%, a 75.17% precision rate, a 91.35% recall rate, an F1 score of 0.8229, and an AUC of 0.8719. Logistic regression achieved an accuracy of 75.59%, a 75.17% precision rate, a 91.35% recall rate, an F1 score of 0.7131, and an AUC of 0.8719. Naïve Bayes achieved an accuracy of 75.59%, a 75.17% precision rate, a 91.35% recall rate, an F1 score of 0.7279, and an AUC of 0.8719 (Figure 5).

## 4. Discussion

Pain is a complex nociceptive stimulus that activates brain regions and networks. Among these brain regions and networks, the PFC plays a vital role in perceiving, modulating, and reappraising pain through various ascending and descending tracts [37]. As proposed by Garcia-Larrea and Peyron [38], the PFC receives encoded pain-related information from the thalamus, incurring attentional and cognitive modulation. Moreover, the PFC is able to reappraise pain-related information and suppressively or facilitatively regulates pain stimuli based on individual psychological and emotional factors. However, for chronic pain patients, the ascending and descending pain pathways are abnormally activated by long-lasting (more than six months) aberrant pain-related information, inducing peripheral and/or central sensitization [39]. Therefore, alterations in brain morphometry and functional connectivity are detected in chronic patients. In particular, the disrupted functional connectivity of the PFC has been associated with chronic patients’ deficits in decision making [40], fear avoidance [41], and working memory [42].

The current study investigated the resting-state brain activity in CP patients compared to HCs using the ALFF as a measure. The results showed significantly increased zALFF values in channel 8, mainly located in the DLPFC, in CP patients compared to HCs. This finding suggests that CP patients may exhibit hyperactivity in the DLPFC, which is consistent with previous studies that have reported altered prefrontal activity in chronic pain conditions [29]. The DLPFC is a region of the brain that is structurally and functionally diverse and plays a crucial role in several brain networks involved in sensory, affective, and cognitive processing. Experimental pain studies consistently reveal DLPFC activation, while CP populations exhibit abnormal increases in DLPFC function, indicating its significance in pain-processing activities such as encoding and modulating acute pain [9,10]. As research has consistently demonstrated the association between chronic pain and structural and functional changes in the DLPFC, this brain region may hold potential as a therapeutic target for pain management. Non-invasive brain stimulation techniques have been shown to effectively alleviate both acute and chronic pain through modulation of the DLPFC, offering promise for future treatment options [43,44].

Interestingly, our findings demonstrated that the CP patients exhibited abnormal emotional responses and were hypersensitive to pain [45], leading to the further worsening of symptoms. Our study also demonstrated that CP patients were associated with higher small-worldness properties and distinct global and local topographic properties of the PFC functional networks as compared to HCs. Consistent with previous findings [46,47], the CP group presented significantly lower global efficiency and synchronizations than the HCs. Synchronizations and global efficiency are reported to be measures of the network information transmission rate, which reflects the PFC’s capacity for information exchange and resource utilization underlying the concurrent processing of information [48,49]. We detected higher small-worldness properties corresponding to the CP patients due to the fact that nodes of processing long-lasting aberrant pain-related information are more likely to communicate together [50,51]. The change in topographic properties might be an important predictive or prognostic biomarker for the identification of the CP phenotype. The normalization of global and local topographic properties can be a valid marker with which to note the efficacy of treatment.

Three supervised machine learning classifiers, namely, linear SVM, logistic regression, and naïve bayes, were used to classify CP patients and HCs, achieving at least 75% accuracy and an F1 score of 0.8229 according to the features from the global and local graph measurements. A previous study [52] used fNIRS and machine learning to identify feelings of different types of pain in HCs, thereby advancing the assessment of pain. Numerous studies have been carried out to determine biomarkers enabling the automatic diagnosis of CP by combing MRI with machine learning using data concerning brain structure [53,54,55], resting state [15,56,57,58], or obtained during the completion of a task [59], with accuracy ranging from 63% to 86%. Compared to fMRI and EEG, the cheap, convenient, and less contraindicatory characteristics of resting-state fNIRS data make them more likely to be widely used in realistic clinical settings, thus benefiting CP patients.

Our study has several limitations. Firstly, it included a limited sample size, which can limit the generalizability of the results. Secondly, the fNIRS technique used in the study has a limited number of channels, which made it difficult to perform regions-of-interest (ROI)-based analysis. Thirdly, the study did not measure the participants’ emotional responses using validated clinical scales, which made it difficult to establish the functional connectivity between the brain regions and the clinical scales.

## 5. Conclusions

In this study, we discovered significantly weaker functional connectivity and notably higher small-worldness properties of resting-state brain networks in the PFC of the CP groups compared to those of the HCs. The detected alternations in macroscopic PFC organization in the chronic pain patients support our central hypothesis that chronic pain can be characterized according to the type of network disorder, which is defined as altered network organization and connectivity. Additionally, the resting-state fNIRS measurements exhibited excellent performance as screening tools for automatically diagnosing chronic pain patients via supervised machine learning, SVM, logistic regression, and naïve bayes, yielding an F1 score of 0.8229 with only seven features.

## Figures and Tables

**Figure 1 bioengineering-10-00669-f001:**
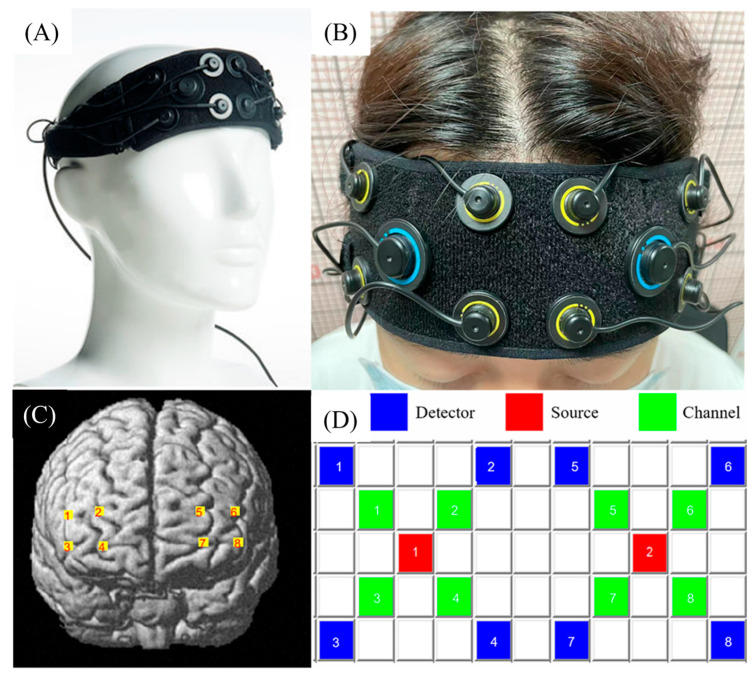
The Artinis device and channel position. (**A**) The Artinis device, composed of 2 near-infrared light source emitters and 8 detectors; (**B**) the device’s setup; (**C**) fNIRS channels reconstructed using NIRS-SPM; (**D**) fNIRS topomaps.

**Figure 2 bioengineering-10-00669-f002:**
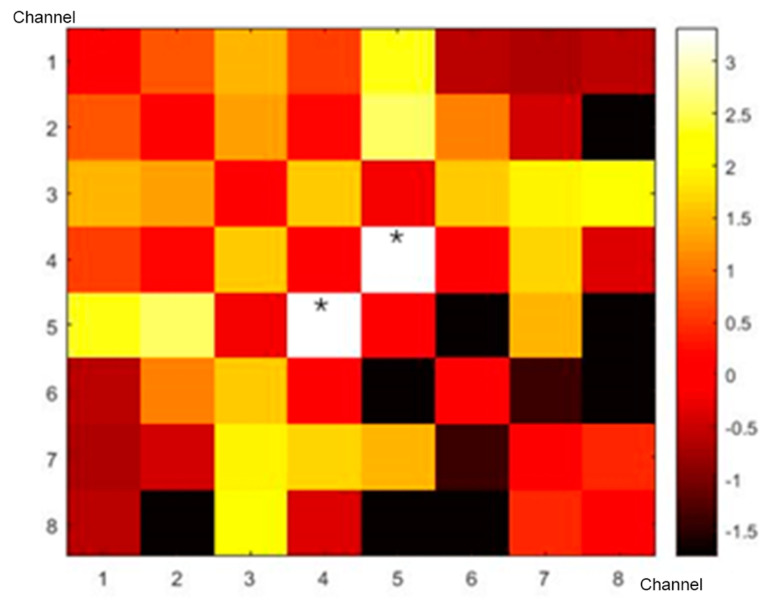
Connectivity difference matrices of HCs and CP patients. Colored bars represent the *t* values of comparison results, and numbers near the axes represent the channels of fNIRS; *, p_FDR_  <  0.05.

**Figure 3 bioengineering-10-00669-f003:**
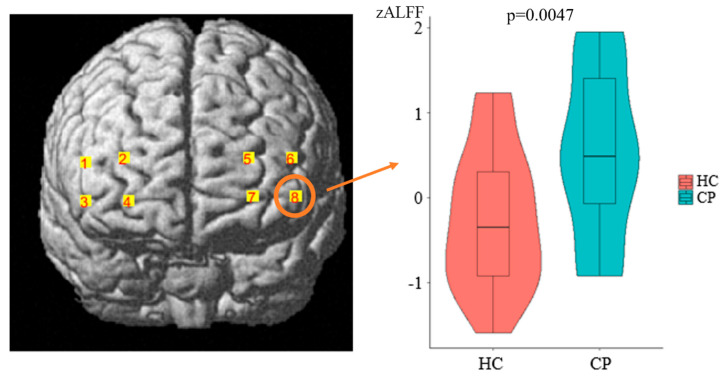
Group comparison of zALFF values between CP and HC groups. Channel 8 exhibited increased zALFF values in CP group as compared to that from HC group. The boxes in the graph represent the interquartile range (IQR) of the data, with the horizontal line inside the box representing the median. The whiskers extending from the box represent the range of the data, excluding any outliers. The shaded areas inside the violin plots represent the density of the data distribution.

**Figure 4 bioengineering-10-00669-f004:**
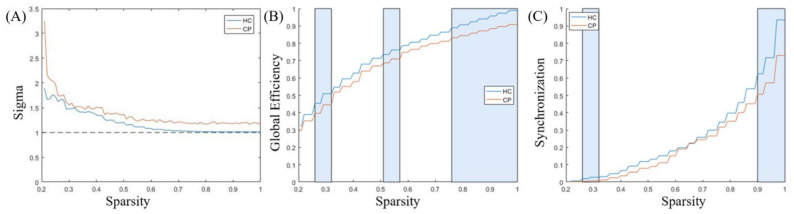
Functional network characteristics of the HC and CP groups. Panel (**A**) represents the measure of small-worldness, and the dashed curve is 1. Panel (**B**) displays the global efficiency, while Panel (**C**) presents the synchronizations. The curves show the network indicators under different thresholds (from 0.21 to 1), in which blue represents the HCs while orange denotes the CP patient group. The rectangles in the background denote the significant differences of the functional network characteristics (*p* < 0.05).

**Figure 5 bioengineering-10-00669-f005:**
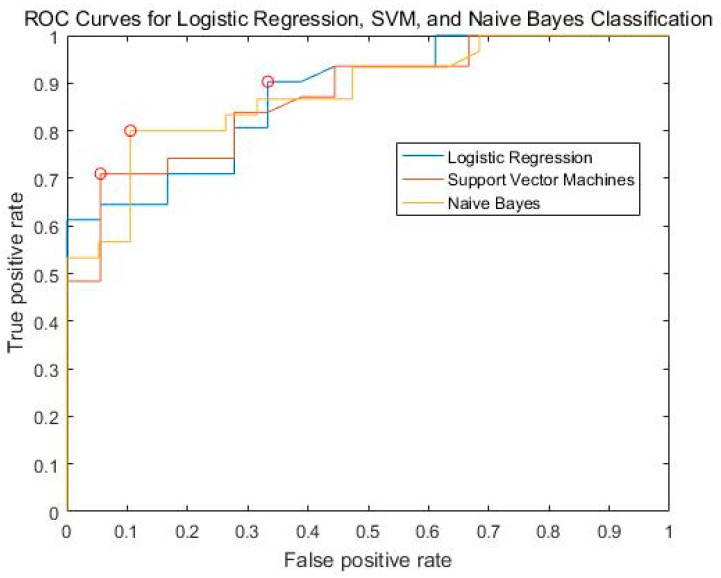
ROC curves for logistic regression, SVM, and naïve bayes classification represented by blue, orange, and yellow colors, respectively. The red circle represented the optimal operating point on the ROC curve. The optimal operating point is determined based on various criteria such as maximizing sensitivity and specificity or minimizing the distance to the ideal point on the curve.

**Table 1 bioengineering-10-00669-t001:** The 3D MNI coordinates, anatomical labels, and coverage percentage of fNIRS channels.

Channel Numbers	MNI			Anatomical Label	Percentage of Overlap
	X	Y	Z		
1	51	46.33	16.33	45—pars triangularis Broca’s area; 46—Dorsolateral prefrontal cortex;	60.517% 39.483%
2	31.67	65	17	10—Frontopolar area; 46—Dorsolateral prefrontal cortex;	80.989% 19.011%
3	50.33	51.67	−1.33	45—pars triangularis Broca’s area; 46—Dorsolateral prefrontal cortex; 47—Inferior prefrontal gyrus;	2.9197% 96.35% 0.72993%
4	30.67	68.33	−1.67	10—Frontopolar area; 11—Orbitofrontal area;	36.093% 63.907%
5	−25.67	66.67	17.67	10—Frontopolar area; 46—Dorsolateral prefrontal cortex;	86.716% 13.284%
6	−46	49.33	17.33	45—pars triangularis Broca’s area; 46—Dorsolateral prefrontal cortex;	41.985% 58.015%
7	−27.67	67.33	0.67	10—Frontopolar area; 11—Orbitofrontal area;	51.495% 48.505%
8	−47.67	51.67	−0.67	10—Frontopolar area; 45—pars triangularis Broca’s area; 46—Dorsolateral prefrontal cortex;	2.2642% 3.7736% 93.962%

**Table 2 bioengineering-10-00669-t002:** The selected features with the highest Fisher scores.

Selected Measurement	HC (Mean ± Standard Deviation)	Pain Patients (Mean ± Standard Deviation)	*t* Value	*p* Values
Network Efficiency	0.6596 ± 0.1093	0.7528 ± 0.0855	−3.1684	0.0027
Nodal Local Efficiency_5	0.5703 ± 0.3846	0.8632 ± 0.2392	−2.9788	0.0045
Nodal Cluster Efficiency 5	0.5129 ± 0.3675	0.8035 ± 0.2646	−2.9981	0.0043
Local Efficiency of Nodal 8	0.6031 ± 0.4249	0.8784 ± 0.1535	−2.7089	0.0093
Community Index of Nodal 7	1.7419 ± 0.6308	1.1053 ± 0.8753	2.9849	0.0045
Clustering coefficient	0.5920 ± 0.1174	0.6829 ± 0.1024	−2.7854	0.0076
Efficiency of Nodal 7	0.6859 ± 0.1328	0.4743 ± 0.3383	3.1267	0.0030

**Table 3 bioengineering-10-00669-t003:** The performance of three classifiers, namely, SVM linear, logistic regression, and naïve bayes.

Learning Model	Accuracy	Precise	Recall	F1 Score	AUC
SVM (linear)	0.7559	0.7517	0.9135	0.8229	0.8719
Logistic Regression	0.7898	0.7418	0.9135	0.7131	0.8754
naïve Bayes	0.7755	0.7297	0.7269	0.7279	0.8781

## Data Availability

The datasets generated for this study are available on request made to the corresponding author.

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
