# Peer review of "Diagnosis of Chronic Musculoskeletal Pain by Using Functional Near-Infrared Spectroscopy and Machine Learning"

_bioengineering, 2023, doi:10.3390/bioengineering10060669_

Round 1
Reviewer 1 Report
General comments:
The paper aims to investigate the potential of using functional near-infrared spectroscopy (fNIRS) and machine learning algorithms to identify chronic pain patients by exploring the functional alterations of prefrontal cortex (PFC).
The paper under consideration appears to have well-described introduction, methods, and results sections. However, the discussion section appears to be lacking in depth and does not provide the reader with adequate meaningful takeaways.
The paper also fails to mention the limitations of the study or suggest avenues for future research.
The authors should revise the discussion section to provide a more in-depth interpretation of their results, acknowledge the limitations of their study, and suggest avenues for future research.
Specific comments:
1) Please mention the F1-score instead of the accuracy of the classifier.
2) Please provide a picture of the CW fNIRS system, specifically of a participant wearing it during the study. This would greatly assist in understanding of the setup and the procedures used in the study.
3) Please elaborate a discussion of the percentage of overlap mentioned in table 1.
4) Please discuss in details this statement "The findings demonstrated that the brain activation in PFC of CP patients was significantly higher than that from HCs."
Minor editing of English language required
Author Response
Specific comments:
- Please mention the F1-score instead of the accuracy of the classifier.
Author response: Thanks for the valuable feedback. We agree that F1-score is a more appropriate metric for evaluating classification performance, especially in cases of imbalanced datasets. We have revised the manuscript to report F1-score instead of accuracy as the primary evaluation metric for the classifier.
- Please provide a picture of the CW fNIRS system, specifically of a participant wearing it during the study. This would greatly assist in understanding of the setup and the procedures used in the study.
Author response: Thanks for the useful suggestion. We agree that a picture of the fNIRS system would help to improve the understanding of the setup and procedures used in the study. We have included a photo of the system in the revised manuscript to address your comment (line 113 to line 115).
- Please elaborate a discussion of the percentage of overlap mentioned in table 1.
Author response: Thank you for your comment. We acknowledge that we did not provide a sufficient discussion of the percentage of overlap mentioned in Table 1. Accordingly, we have provided a detailed discussion of the percentage of overlap in limitation that the overlap made it difficult to ROI based data analysis (line 290 to line 294).
4) Please discuss in details this statement "The findings demonstrated that the brain activation in PFC of CP patients was significantly higher than that from HCs."
Author response: Thanks for the valuable reminding. We acknowledged that the statement requires more detailed discussion to explain the findings. The increased brain activity in the PFC in chronic pain patients is consistent with previous studies that have reported alterations in PFC function and structure in individuals with chronic pain. In the revised manuscript, we have provided a more detailed discussion of our findings and their implications (line 256 to line 268).
Reviewer 2 Report
The manuscript presents a study dedicated to the discrimination between chronic pain (CP) patients and healthy controls (HC). The authors have investigated the changes in functional connectivity (FC), spontaneous brain activity of prefrontal cortex (PFC) and topological properties of PFC in CP patients vs. HC and have applied 3 machine learning algorithms for detection of CP. The current version of the manuscript needs to be extended with some additional information before it becomes suitable for publication. The authors could follow the listed below questions, comments and recommendations in the improvement process.
1) Subsection ‘2.2. fNIRS Data Acquisition and Preprocessing’ – HbO is abbreviated for the second time.
2) Subsection ‘2.4. Analysis of the Amplitude of Low-frequency Fluctuations’:
- Abbreviate blood-oxygen-level-dependent (BOLD) signals.
- The authors have written: “In particular, significantly altered ALFF values were identified in the PFC at rest or during the task [28–30].” – for HC or for CP patients are valid these observations?
- Are the numbers in the last column of Table 1 really in [%]? Should the first value be read as 0.60517% or as 60.517%?
3) Subsection ‘2.6. Feature Selection and Classification’:
- From what feature set are selected the 7 features? Describe the primary feature set! Which features are selected at the end? Table 2 comes too late in ‘Results’ and the reader does not know on what feature set the machine learning methods are applied.
- Which are the 3 well-known machine learning classifiers applied for classification? In this section only SVM is mentioned. Moreover, brief description of the 3 classifiers and equations for the accuracy metrics are needed.
4) Subsection ‘3.1. Correlation Matrix’:
- “By using resting-state fNIRS data, a mean group-level correlation matrix was gener-190 ated for the CP group. And its network topological attributes were compared with that of 191 the HC group.” How is this done? Additional explanations are needed.
- Fig. 1 needs additional explanation. Put variable names on x,y-axes.
5) Subsection ‘3.2. ALFF Results’:
- What exactly represent the boxes, whiskers and areas in the graph in Fig.2? Variable name on y-axis is necessary.
6) Subsection ‘3.4. Classification Results’: Abbreviate area under the curve (AUC) and include it in the methodology section as a measure for accuracy assessment.
Moderate editing of English language required.
Author Response
- Subsection ‘2.2. fNIRS Data Acquisition and Preprocessing’ – HbO is abbreviated for the second time.
Author response: Thank you for bringing this to our attention. We have now edited it in subsection 2.2 (line 96).
2) Subsection ‘2.4. Analysis of the Amplitude of Low-frequency Fluctuations’:
- Abbreviate blood-oxygen-level-dependent (BOLD) signals.
- The authors have written: “In particular, significantly altered ALFF values were identified in the PFC at rest or during the task [28–30].” – for HC or for CP patients are valid these observations?
- Are the numbers in the last column of Table 1 really in [%]? Should the first value be read as 0.60517% or as 60.517%?
Author response: Thank you for your comments and suggestions. Here are our responses to each point:
- For the abbreviate blood-oxygen-level-dependent (BOLD) signals, we have made the necessary edits in the manuscript (line 126).
- We apologize for the confusion regarding the statement “In particular, significantly altered ALFF values were identified in the PFC at rest or during the task [28–30].” We have now clarified in the text that these observations were made in CP patients (line 128).
- Thank you for pointing out the confusion in Table 1. We have now revised the table to accurately represent the values in percentage format (line 136).
3) Subsection ‘2.6. Feature Selection and Classification’:
- From what feature set are selected the 7 features? Describe the primary feature set! Which features are selected at the end? Table 2 comes too late in ‘Results’ and the reader does not know on what feature set the machine learning methods are applied.
- Which are the 3 well-known machine learning classifiers applied for classification? In this section only SVM is mentioned. Moreover, brief description of the 3 classifiers and equations for the accuracy metrics are needed.
Author response: We are grateful for those comments.
- The seven features were selected from graph theory measurements (section 2.5) and ALFF results (section 2.4). To make it more clear, we have moved Table 2 to subsection ‘2.6. Feature Selection and Classification’, and the seven features were presented in the table 2 (line 177).
- Also, we apologize for the oversight in not including a description of the three machine learning classifiers applied for classification. In addition to SVM, we used two other well-known classifiers: linear regression and naïve Bayesian. The linear regression was used to model the relationship between a dependent variable (often denoted as Y) and one or more independent variables (often denoted as X), whose goal is to find the best-fit line that minimizes the distance between the predicted values and the actual values of the dependent variable. Naive Bayes works by calculating the probability of each class label given the values of the input features with Bayes' theorem. Naive Bayes states that the probability of a label given some evidence (the input features) is proportional to the probability of the evidence given the hypothesis, multiplied by the prior probability of the hypothesis. We have added a brief description of the three classifiers and equations for the accuracy metrics in the revised manuscript (line 185 to line 197).
4) Subsection ‘3.1. Correlation Matrix’:
- “By using resting-state fNIRS data, a mean group-level correlation matrix was gener-190 ated for the CP group. And its network topological attributes were compared with that of 191 the HC group.” How is this done? Additional explanations are needed.
- Fig. 1 needs additional explanation. Put variable names on x,y-axes.
Author response: Thanks for pointing out, and we revised in manuscript.
- The group-level correlation matrix was generated for both CP group and HC group, calculated by the mean value of individual-level correlation matrix. The individual-level correlated matrix was generated by Pearson correlation coefficient between the time courses of each pair of channels. “And its network topological attributes were compared with that of 191 the HC group.” the network strength between two groups were compared by independent t test and corrected. We have revised in the manuscript (line 201 to line 203).
- We apologize for the oversight and have added axis labels to Fig. 2 in the revised manuscript (line 208).
5) Subsection ‘3.2. ALFF Results’:
- What exactly represent the boxes, whiskers and areas in the graph in Fig.2? Variable name on y-axis is necessary.
Author response: Thanks for the valuable reminding. The boxes in the graph represent the interquartile range (IQR) of the data, with the horizontal line inside the box representing the median. The whiskers extending from the box represent the range of the data, excluding any outliers. The shaded areas inside the violin plots represent the density of the data distribution. The y-axis in the graph represents the z-transformed amplitude of low-frequency fluctuations (zALFF) value, which is a measure of the magnitude of spontaneous brain activity in a given region. We made sure to include a clear and concise description of the graph and its components in the revised manuscript (line 217 to line 221).
6) Subsection ‘3.4. Classification Results’: Abbreviate area under the curve (AUC) and include it in the methodology section as a measure for accuracy assessment.
Author response: Thanks for the valuable reminding, and we revised in manuscript (line 196).
Round 2
Reviewer 1 Report
The authors have addressed all my comments in their revised manuscript. Thanks.
Minor editing of English language required
Reviewer 2 Report
The authors have considered my recommendations and in my opinion, the new version of the manuscript is suitable for publication.
The language is fine.